# Clinical Utility of Rapid On-Site Evaluation of Touch Imprint Cytology during Cryobiopsy for Peripheral Pulmonary Lesions

**DOI:** 10.3390/cancers14184493

**Published:** 2022-09-16

**Authors:** Yutaka Muto, Keigo Uchimura, Tatsuya Imabayashi, Yuji Matsumoto, Hideaki Furuse, Takaaki Tsuchida

**Affiliations:** 1Department of Endoscopy, Respiratory Endoscopy Division, National Cancer Center Hospital, 5-1-1 Tsukiji Chuo-ku, Tokyo 104-0045, Japan; 2Department of Respiratory Medicine, Japanese Red Cross Medical Center, 4-1-22 Hiroo Shibuya-ku, Tokyo 150-8953, Japan

**Keywords:** lung cancer, bronchoscopy, cryobiopsy, diagnosis, rapid on-site evaluation, touch imprint cytology

## Abstract

**Simple Summary:**

With increasing interest in precision medicine for lung cancer, cryobiopsy is expected to improve the success rate not only for histological diagnosis, but also for next-generation sequencing. Rapid on-site evaluation (ROSE) is an immediate cytological evaluation performed during bronchoscopy. However, little is known about its clinical utility during cryobiopsy. We retrospectively reviewed the data of 63 consecutive patients who underwent cryobiopsy with ROSE of touch imprint cytology (ROSE-TIC) for solid peripheral pulmonary lesions. When the results of ROSE-TIC of each patient were compared directly with the histological findings of the corresponding specimen, the sensitivity, specificity, and positive and negative predictive values were 69.8%, 90.0%, 93.8%, and 58.1%, respectively. The concordance rate was 76.2%. Therefore, we believe that ROSE-TIC, due to its high specificity and positive predictive value, may be a potential tool in deciding whether cryobiopsy sampling could be finished during bronchoscopy.

**Abstract:**

Cryobiopsy enables us to obtain larger specimens than conventional forceps biopsy despite the caution regarding complications. This study aimed to evaluate the clinical utility of rapid on-site evaluation of touch imprint cytology (ROSE-TIC) during cryobiopsy of peripheral pulmonary lesions (PPLs). We retrospectively reviewed the data of consecutive patients who underwent cryobiopsy for solid PPLs between June 2020 and December 2021. ROSE-TIC was performed on the first specimen obtained via cryobiopsy and assessed using Diff-Quik staining. The results of ROSE-TIC for each patient were compared with the histological findings of the first cryobiopsy specimen. Sixty-three patients were enrolled in this study. Overall, 57 (90.5%) lesions were ≤30 mm in size and 37 (58.7%) had positive bronchus signs. The radial endobronchial ultrasound findings were located within and adjacent to the lesion in 46.0% and 54.0% of the cases, respectively. The sensitivity, specificity, and positive and negative predictive values of the ROSE results for histological findings of the corresponding specimens were 69.8%, 90.0%, 93.8%, and 58.1%, respectively. The concordance rate was 76.2%. In conclusion, ROSE-TIC, due to its high specificity and positive predictive value, may be a potential tool in deciding whether cryobiopsy sampling could be finished during bronchoscopy.

## 1. Introduction

Lung cancer is one of the most commonly diagnosed cancers and the leading cause of cancer-related deaths worldwide [1]. Based on the results of several randomized trials [2,3], lung cancer screening using low-dose computed tomography (CT) is recommended for high-risk patients, such as those aged ≥50 years with a ≥20 pack-year history of smoking [4]. Following the adoption of screening, the detection of peripheral pulmonary lesions (PPLs) has increased. Although transbronchial forceps biopsy is one of the relatively noninvasive and widely used strategies for diagnosing PPLs, histological diagnosis can be challenging [5,6].

The cryoprobe enables us to obtain a relatively large tissue sample in a 360° manner with a minimized crush artifact [7]. With increasing interest in precision medicine for lung cancer, cryobiopsy is expected to improve the success rate not only for histological diagnosis, but also for next-generation sequencing. A randomized control trial reported that cryobiopsy did not show a statistically significant difference in diagnostic yield compared with forceps biopsy (74.2% with cryobiopsy vs. 60.5% with forceps biopsy, *p* = 0.42) [8]; however, several studies have shown that cryobiopsy in combination with other conventional sampling methods increased the diagnostic yield for PPLs [7,9]. In addition, the cryoprobe can obtain significantly larger amounts of both DNA and RNA extracted from samples than the forceps [10]. Although several studies have supported the safety of cryobiopsy for PPLs [7,9,11,12], bleeding is a common complication, making repeated biopsies difficult for bronchoscopists.

Rapid on-site evaluation (ROSE) is an immediate cytological evaluation during bronchoscopy that helps in intraprocedural decision-making. Its clinical utility during endobronchial ultrasound-guided transbronchial needle aspiration (EBUS-TBNA) has been reported in a randomized controlled study [13], and the CHEST guideline states that ROSE may reduce the number of aspirations and other procedures required, whereas it does not affect the diagnostic yield [14]. Regarding PPLs, some studies reported that ROSE using biopsy specimens reduced the number of biopsies and complication rates in transbronchial forceps biopsy [15,16]. Therefore, we presumed that ROSE has the potential to reduce the number of cryobiopsies. A small single-arm retrospective study showed that the diagnostic accuracy of ROSE of touch imprint cytology (ROSE-TIC) for final diagnosis was 73.9% during cryobiopsy for PPLs [17]. However, to the best of our knowledge, no study has compared the results of ROSE-TIC with the corresponding cryobiopsy specimens.

Hence, this study aimed to evaluate the clinical utility and feasibility of ROSE-TIC during cryobiopsy of PPLs. We modified the method of TIC from a previous study [17] and compared the results of ROSE-TIC directly with the histological findings of the corresponding cryobiopsy specimens.

## 2. Materials and Methods

### 2.1. Eligibility Criteria

This retrospective study enrolled all consecutive patients who underwent cryobiopsy for solid PPLs, using a 1.7 mm single-use cryoprobe (20402-410, ERBE, Tubingen, Germany) and ERBECRYO^®^2 (ERBE, Tubingen, Germany) at the National Cancer Center Hospital, Tokyo, Japan, between June 2020 and December 2021. Patients who did not undergo ROSE-TIC using cryobiopsy specimens or whose target lesions were bronchoscopically visible were excluded.

### 2.2. Demographics and Radiological Findings

Data on age and sex were obtained from the patients’ electronic medical records. The lesion size, involved lobe/segment, bronchus sign, and distance from the costal pleura were evaluated using high-resolution CT (HRCT) before bronchoscopy. A positive bronchus sign was defined as the presence of a bronchus in the cross-section, leading to or contained within the nodule or mass [18]. The target bronchus leading to the lesion suitable for biopsy was planned using virtual bronchoscopic navigation (Ziostation2; Ziosoft Ltd., Tokyo, Japan).

### 2.3. Biopsy Procedures

Bronchoscopy was performed using flexible bronchoscopes with a radial endobronchial ultrasound (R-EBUS) probe (UM-S20-17 S; Olympus, Tokyo, Japan) and X-ray fluoroscopy. The patients were moderately or deeply sedated using intravenous anesthesia with opioids and sedatives (midazolam or propofol) in combination with local anesthesia (2% lidocaine intratracheally) [19]. The patients were endotracheally intubated to secure the airway.

Before starting the bronchoscopy, the visibility of the lesion on radiography was checked. We applied the two-scope technique when we performed the cryobiopsy [11,20]; one thin scope (P260F or P290; Olympus, Tokyo, Japan) was used for diagnostic biopsy, the other (therapeutic) scope (1T260 or 1TQ290; Olympus, Tokyo, Japan)—for bleeding control. Using the thin scope, the target bronchus was approached. The R-EBUS probe was inserted via the working channel of the thin scope into the pre-planned target bronchus. The target bronchus was determined based on X-ray fluoroscopy and R-EBUS findings. From the position of R-EBUS to the lesion, the findings on R-EBUS were classified as follows: “within”, “adjacent to”, and “invisible” [21]. The lesions were frozen for 3–5 s each using a 1.7-mm single-use cryoprobe. Upon removal of the thin scope with the cryoprobe, the assistant bronchoscopist immediately inserted the therapeutic scope and fixed it into the involved bronchus to control bleeding after each cryobiopsy. The suggested number of specimens was three, with a maximum limit of four [22]. The periods of time required for the first cryobiopsy and for the entire procedure were separately recorded.

Bleeding complications and postinterventional pneumothorax were recorded. The severity of bleeding was reported based on the standardized airway bleeding scale from a Delphi consensus statement from the Nashville Working Group: mild bleeding, blood suctioning required for <1 min; moderate bleeding, suctioning required for >1 min; repeat wedging of the bronchoscope for persistent bleeding needed; application of cold saline, diluted vasoactive substance, or thrombin; severe bleeding, selective intubation needed for <20 min; life-threatening bleeding, persistent selective intubation or emergency care required [23].

### 2.4. Specimen Processing

The first specimen obtained by cryobiopsy was immediately imprinted onto two slide glasses. The specimens attached to the cryoprobe were not thawed in saline but rotated and stamped such that the entire surface of the spherical specimen was attached to the slide glasses (Figure 1). One slide was rapidly air-dried and stained with modified Giemsa (Diff-Quik; Sysmex Ltd., Kobe, Japan) for ROSE, whereas the other was fixed in 95% alcohol for Papanicolaou staining. The remaining specimens were fixed with 10% formalin and embedded in paraffin. The specimens stained with modified Giemsa were immediately evaluated in the same bronchoscopy room by a bronchoscopist with sufficient experience (at least five years with at least 200 cases per year) of pulmonary cytopathology (T.I. or T.T.) [24,25]. The outcome of ROSE was judged either positive or negative for malignancy, and the cytology specialist immediately informed the attending bronchoscopists of the result. The Papanicolaou-stained specimens were evaluated by cytotechnologists and cytology specialists.

### 2.5. Histological Diagnosis by Cryobiopsy and Final Diagnosis

When cryobiopsies were performed several times, the specimens were separately fixed in 10% formalin and embedded in paraffin. Each specimen was evaluated by the same pathologists. The final diagnosis was made based on bronchoscopic diagnosis or surgery, if conducted. When the diagnosis could not be confirmed on bronchoscopy and the lesion size did not increase for at least 6 months after the procedure, the lesion was regarded as benign. Clinical data were monitored until 30 June 2022.

### 2.6. Statistical Analysis

Categorical variables were presented as numbers with percentages, continuous variables—as medians with ranges. The sensitivity, specificity, positive predictive value (PPV), and negative predictive value of the histological diagnosis of cryobiopsy to the final diagnosis were calculated. Those of the results of ROSE-TIC to the histological findings of the first cryobiopsy specimens were also calculated. All the statistical analyses were performed using EZR software (Saitama Medical Center, Jichi Medical University, Saitama, Japan). Statistical significance was set at a two-tailed *p*-value < 0.05.

### 2.7. Ethics Statement

Written informed consent for bronchoscopy was obtained from all the patients. As this was a retrospective study, the need for additional informed consent was waived. This study was approved by the institutional review board of the National Cancer Center Hospital, Tokyo, Japan (No. 2018-090).

## 3. Results

### 3.1. Patient Characteristics

A total of 172 patients underwent cryobiopsy using a 1.7 mm single-use cryoprobe for PPLs at our hospital from June 2020 to December 2021. Finally, 63 patients were included in this study (Figure 2).

Table 1 shows the patient characteristics in this study. The median age was 70 years (range, 42–87 years), and 40 patients (63.5%) were male. The median lesion size was 20.6 (4.9–86.7) mm, and 29 (46.0%) lesions were located in the right upper lobe or left upper segment. The bronchus sign was positive in 37 (58.7%) cases, and the median distance from the costal pleura was 8.1 (0–42.2) mm on the preprocedural HRCT scan.

On X-ray fluoroscopy, 57 lesions (90.5%) were visible. R-EBUS images showed that 29 (46.0%) and 34 (54.0%) cases were classified as “within” and “adjacent to” the lesion, respectively. The median number of specimens obtained using cryobiopsy was three (range, 1–4). None of the patients underwent conventional forceps biopsy or needle aspiration before or after the cryobiopsy.

Severe bleeding occurred in one patient (1.6%), and there were no life-threatening bleeding complications. Surgery for lung nodules or masses was performed in 33 patients (52.4%) during the study period. Finally, 60 lesions (95.2%) were confirmed to be malignant. Adenocarcinoma was the most common lesion, diagnosed in 32 lesions (50.8%).

### 3.2. Correlation between the Histological Diagnosis Using Cryobiopsy Specimens and the Final Diagnosis

In total, 52 lesions (82.5%) were diagnosed as malignant using cryobiopsy. Histological diagnosis of the cryobiopsy specimens yielded sensitivity, specificity, and positive and negative predictive values of 86.7%, 100%, 100%, and 27.3%, respectively, for the diagnosis of malignant lesions. The diagnostic accuracy was 87.3%. Of the 52 cases, malignant cells were detected in the first cryobiopsy specimen in 43 (82.7%).

### 3.3. Correlation between ROSE-TIC and the Histological Findings of the First Specimen Obtained Using Cryobiopsy

In total, 32 specimens (50.8%) were judged to be positive for malignancy in ROSE-TIC. Table 2 shows a comparison of the results of ROSE-TIC with the histological findings of the first specimens obtained using cryobiopsy. The sensitivity, specificity, and positive and negative predictive values of the ROSE-TIC results for histological findings of the corresponding cryobiopsy specimens were 69.8%, 90.0%, 93.8%, and 58.1%, respectively. The concordance rate was 76.2%. There were two false-positive cases in the present study. They were eventually diagnosed as adenocarcinoma and lymphoepithelial carcinoma, respectively, using surgical specimens. In one case of adenocarcinoma, an evaluator misjudged degenerative bronchial cells surrounding the lesions as adenocarcinoma. In the other case of lymphoepithelial carcinoma, the evaluator mistakenly described an alveolar epithelium with reactive changes as an adenocarcinoma.

### 3.4. Correlation between ROSE-TIC and the Cytological Results of the First Specimen Obtained Using Cryobiopsy

In total, 38 specimens (60.3%) were judged to be positive for malignancy in cytology. Table 3 shows a comparison of the results of ROSE-TIC with the cytological results of the first specimen obtained using cryobiopsy. The sensitivity, specificity, and positive and negative predictive values of the ROSE-TIC results for the cytological results of the corresponding cryobiopsy specimens were 78.9%, 92.0%, 93.8%, and 74.2%, respectively. The concordance rate was 84.1%.

## 4. Discussion

We examined the clinical utility and feasibility of ROSE-TIC in cryobiopsy specimens of solid PPLs suspected to be lung cancer. To the best of our knowledge, this was the first report comparing the results of ROSE-TIC directly with the histological findings of the corresponding cryobiopsy specimens. This study demonstrated the high specificity and PPV of ROSE-TIC during cryobiopsy.

In the present study, the results of ROSE-TIC were compared with the histological findings of the corresponding specimens. The utility of ROSE has been established in EBUS-TBNA and is recommended in the American College of Chest Physicians guidelines [14]. In the context of forceps biopsy for PPLs, several studies showed that the sensitivity, specificity, and concordance rate of ROSE for bronchoscopic diagnosis or final diagnosis were 88.2–97.7%, 65.9–100%, and 85.2–98.3%, respectively [25,26,27,28,29,30] (Table 4). A retrospective study of 23 patients reported that the sensitivity, specificity, and diagnostic accuracy of ROSE-TIC during cryobiopsy to the final diagnosis were 70%, 100%, and 73.9%, respectively [17]. However, since bronchoscopists would make intraprocedural decisions based on ROSE-TIC results, it is more desirable to compare ROSE-TIC directly with the histological findings of the corresponding cryobiopsy specimen than with the bronchoscopic diagnosis or final diagnosis. In the present study, the sensitivity, specificity, and concordance rate of ROSE-TIC with the corresponding specimen were 69.8%, 90.0%, and 76.2%, respectively, which are similar to those of a previous study that compared ROSE-TIC with the final diagnosis [17]. In addition, as can be seen in Table 4, the sensitivity and concordance rate of ROSE and the pathological diagnosis is lower in cryobiopsy than in forceps biopsy. One possible factor might be the evaluator of ROSE. Although a retrospective study with forceps biopsy specimens suggested that ROSE performed by a trained pulmonologist had a high correlation to the formal cytologic report by cytopathologists [25], to our knowledge, there is no study comparing the results of ROSE-TIC during cryobiopsy performed by trained pulmonologists with those performed by pathologists. 

The high specificity of ROSE-TIC during cryobiopsy indicates that it could potentially reduce the number of biopsies. The results of ROSE-TIC may help bronchoscopists decide whether to add another sample. The major complications of cryobiopsy are bleeding and pneumothorax [31]. Prophylactic balloon occlusion and two-scope techniques have been developed to prevent severe bleeding [11,20,22,32]. Previous studies reported that moderate and severe bleeding occurred in 14.0–38.9% and 0–1.2% of cases, respectively, and the incidence of pneumothorax was 0.8–6.6% when cryobiopsy was performed for PPLs [7,9,11,12]. A high number of cryobiopsies has been reported to be unrelated to the severity of bleeding [33]; however, bleeding obstructs the visual field during bronchoscopy and reduces the approachability of the target bronchus. Although the optimal number of cryobiopsies for PPLs has not been established, avoiding unnecessary cryobiopsies is preferable. Regarding ROSE-TIC during forceps biopsy for PPLs, a randomized prospective study showed that ROSE-TIC shortened the procedure time (mean, 24.6 ± 6.8 min with ROSE vs. 32.4 ± 8.7 min without ROSE; *p* = 0.001), decreased the number of biopsies (2.6 ± 0.4 vs. 4.8 ± 0.2, *p* = 0.021), and improved the diagnostic yield (85.9% vs. 70.3%, *p* = 0.016) [16]. These benefits are expected in ROSE-TIC during cryobiopsy as well.

Our ROSE-TIC method showed a high PPV of 93.4%; however, there were two false-positive cases in the present study. Respiratory cytology has numerous potential pitfalls, and pulmonary lesions are more difficult to categorize by pulmonologists than lymph nodes [34]. For example, alveolar epithelium with reactive changes or degenerative bronchial cells surrounding the lesions sometimes misleads the judgment in the conditions of limited examination time [27,35,36]. In addition, HRCT or R-EBUS findings might have influenced the judgment since ROSE was conducted in the bronchoscopy room, and the ROSE evaluator could have known the imaging findings. Another possible reason for the false-positive results is that malignant cells did not exist on the cut surface of the specimen for histological diagnosis despite the presence of malignant cells outside of the spherical specimen.

There is no established TIC method for cryobiopsy specimens; however, we consider this an important issue. In a recent study on ROSE-TIC during cryobiopsy, Arimura et al. thawed the tissue attached to the tip of the cryoprobe in saline and cut it in half before stamping it [17]. The cut surface with a large number of tumor cells was stamped onto a glass slide using this method when the cryoprobe was within the target lesion on the R-EBUS image. However, this method may not be effective when the cryoprobe is adjacent to the lesion such as in our study, wherein the “adjacent to” classification was noted in more than half the lesions (54%) using R-EBUS. In the present study, the specimens were imprinted onto glass slides without cutting them in half so that the entire surface of the specimen was in contact with the glass slide. A retrospective study reported that TIC without cutting improved the diagnostic yield in combination with histological diagnosis of cryobiopsy [33]. The proposed TIC method has three strengths. First, a large surface area can be stamped onto slide glasses. Assuming that the specimen is spherical, its surface area is four times larger than its cross-sectional area. Although it is difficult to smear the entire surface of the specimen onto a glass slide, our TIC method was expected to collect many cells. Second, cell swelling, which makes ROSE difficult, is prevented because there is no need to thaw cryobiopsy specimens in saline. Third, omitting the process of cutting saves time. Further studies are required to develop a reliable and time-saving method for TIC using cryobiopsy specimens.

This study has some limitations. First, this was a retrospective study conducted at a single cancer center. Thus, the sample size was small and selection bias may have existed. Second, the study had a single-arm design, and we did not compare data of the patients who received ROSE and those who did not to show its relative utility. Third, only solid PPLs were included in this study. Thus, it is unclear whether the sensitivity and specificity of ROSE-TIC during cryobiopsy for ground-glass nodules to the histological results of cryobiopsy specimens would be similar to those for solid PPLs. Fourth, in this study, ROSE-TIC was evaluated only on the first cryobiopsy specimens, without other prior sampling methods. It is unclear whether tissue sampling prior to cryobiopsy would yield the same results as in this study for the second and subsequent cryobiopsies because of the potential impact of bleeding associated with prior tissue sampling on ROSE-TIC. Finally, it is unclear whether our ROSE-TIC results can be applied to other kinds of staining used for ROSE such as Papanicolaou-based rapid staining [24,37]. A larger prospective multi-arm study is required to confirm the clinical benefit of ROSE-TIC during cryobiopsy.

## 5. Conclusions

Although the results of ROSE-TIC performed by bronchoscopists were not consistent with the histological findings of the corresponding cryobiopsy specimens in approximately one fourth of the cases, our study suggests that ROSE-TIC has potential utility in cryobiopsy for solid PPLs. The high specificity and PPV indicate that ROSE-TIC could help bronchoscopists intraprocedurally decide whether cryobiopsy sampling could be finished.

## Figures and Tables

**Figure 1 cancers-14-04493-f001:**
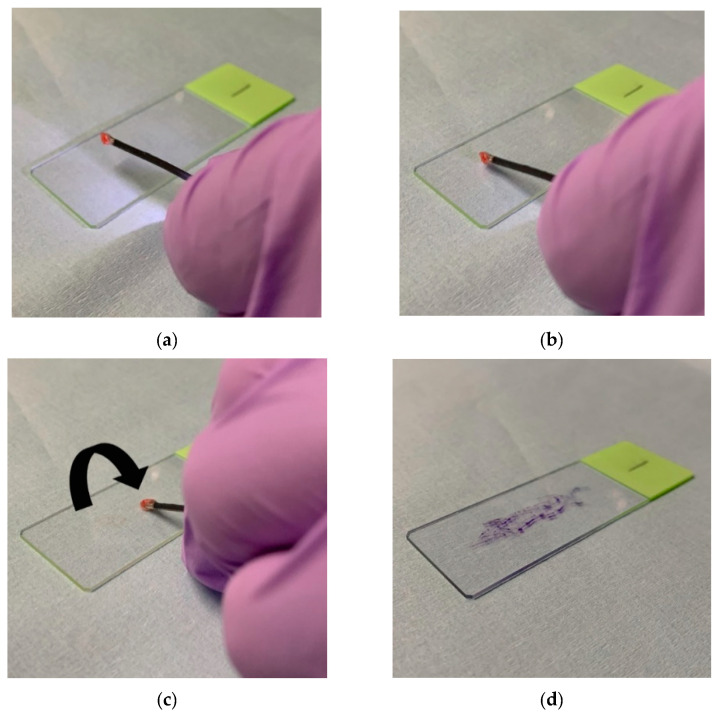
Rapid on-site evaluation of touch imprint cytology used in this study. (**a**) A cryobiopsy specimen is attached to the cryoprobe. (**b**) The specimen is imprinted onto a slide glass without being thawed in saline. (**c**) Then, the specimen is rotated and stamped repeatedly such that the entire surface of the spherical specimen can be attached to the slide glass. (**d**) Finally, the slide glass is stained with modified Giemsa (Diff-Quik; Sysmex Ltd., Kobe, Japan) and evaluated by a bronchoscopist with sufficient experience of pulmonary cytopathology.

**Figure 2 cancers-14-04493-f002:**
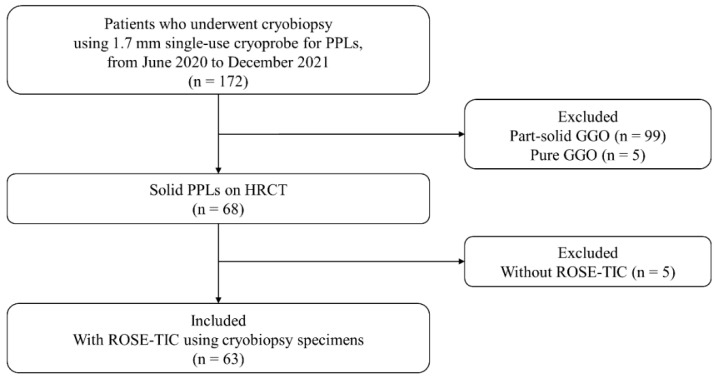
Flowchart of patient recruitment. GGO, ground-glass opacity; HRCT, high-resolution computed tomography; PPL, peripheral pulmonary lesion; ROSE-TIC, rapid on-site evaluation of touch imprint cytology.

**Table 1 cancers-14-04493-t001:** Characteristics of the patients who underwent cryobiopsy and rapid on-site evaluation of touch imprint cytology for solid peripheral pulmonary lesions.

Characteristics	
Age, years	70 (42–87)
Sex	
Male	40 (63.5)
Female	23 (36.5)
Lesion size, mm	20.6 (4.9–86.7)
Involved lobe/segment	
RUL and LUS	29 (46.0)
RML and lingula	12 (19.0)
RLL and LLL	22 (34.9)
Positive bronchus sign	37 (58.7)
Distance from the costal pleura, mm	8.1 (0–42.2)
Fine visibility on radiography	57 (90.5)
Radial EBUS image	
Within	29 (46.0)
Adjacent to	34 (54.0)
Number of specimens	3 (1–4)
Time required for the first biopsy, min	11.1 (6.7–39.3)
Procedure time, min	29.2 (21.7–60.7)
Bleeding complications	
No	4 (6.3)
Mild	23 (36.4)
Moderate	35 (55.6)
Severe	1 (1.6)
Life-threatening	0 (0)
Postinterventional pneumothorax	1 (1.6)
Surgical resection performed	33 (52.4)
Final diagnosis	
Adenocarcinoma	32 (50.8)
Squamous cell carcinoma	11 (17.5)
Lymphoepithelial carcinoma	1 (1.6)
Adenosquamous cell carcinoma	1 (1.6)
Non-small-cell carcinoma	1 (1.6)
Spindle cell carcinoma	1 (1.6)
Small-cell carcinoma	1 (1.6)
Large-cell neuroendocrine carcinoma	1 (1.6)
Metastatic tumor	9 (14.3)
Carcinoid tumor	1 (1.6)
MALT lymphoma	1 (1.6)
Benign	3 (4.8)

The data presented are the number of patients (%) or the median (range). EBUS, endobronchial ultrasound; LUS, left upper segment; LLL, left lower lobe; MALT, mucosa-associated lymphoid tissue; RUL, right upper lobe; RML, right middle lobe; RLL, right lower lobe.

**Table 2 cancers-14-04493-t002:** Correlation between the results of rapid on-site evaluation of touch imprint cytology and the histological findings of the first specimen obtained by cryobiopsy.

ROSE-TIC	Histological Findings of the First Specimen Obtained by Cryobiopsy		
	Malignancy	Nonmalignancy	Total
Positive	30	2	32
Negative	13	18	31
Total	43	20	63

Sensitivity, 69.8%; specificity, 90.0%; positive predictive value, 93.8%; negative predictive value, 58.1%; concordance rate, 76.2%. ROSE-TIC, rapid on-site evaluation of touch imprint cytology.

**Table 3 cancers-14-04493-t003:** Correlation between the results of rapid on-site evaluation of touch imprint cytology and the cytological results of the first specimen obtained by cryobiopsy.

ROSE-TIC	Cytological Results of the First Specimen Obtained by Cryobiopsy		
	Malignancy	Nonmalignancy	Total
Positive	30	2	32
Negative	8	23	31
Total	38	25	63

Sensitivity, 78.9%; specificity, 92.0%; positive predictive value, 93.8%; negative predictive value, 74.2%; concordance rate, 84.1%. ROSE-TIC, rapid on-site evaluation of touch imprint cytology.

**Table 4 cancers-14-04493-t004:** The sensitivity, specificity, and concordance rate of rapid on-site evaluation using forceps biopsy or cryobiopsy specimens for peripheral pulmonary lesions to the pathological diagnosis in the previous studies and the present study.

Study	Design	N	Sampling Methods for ROSE	Evaluator of ROSE	Staining for ROSE	Sensitivity of ROSE	Specificity of ROSE	Concordance Rate of ROSE	Pathology Specimens Compared with ROSE
Lin et al. [25]	Retrospective	86	Biopsy	Trained pulmonologist	Hemacolor	88.2%	80.0%	87.2%	TBB
Chen et al. [26]	Retrospective	279	Brushing or biopsy	Cytopathologist	Rapid Liu	98.2%	100%	98.3%	TBB or brushing
Izumo et al. [27]	Retrospective	718	Brushing or biopsy	Trained pulmonologist	Diff-Quik	88.6%	65.9%	80.1%	TBB or brushing
Maekura et al. [28]	Prospective	45	Brushing, curettage, or biopsy	Cytotechnologist	Ultrafast Papanicolaou	90.6%	92.3%	91.1%	Final diagnosis
Shikano et al. [29]	Retrospective	460 *	Biopsy	Cytotechnologist	Diff-Quik	91.1%	90.4%	90.9%	TBB
Wan et al. [30]	Retrospective	115	Brushing, aspiration, or biopsy	Certificated cytology scientist	Diff-Quik	97.7%	77.8%	85.2%	TBB
Arimura et al. [17]	Retrospective	23	Cryobiopsy	Experienced pathologist	Diff-Quik	70.0%	100%	73.9%	Final diagnosis
Our study	Retrospective	63	Cryobiopsy	Trained pulmonologist	Diff-Quik	69.8%	90.0%	76.2%	Corresponding cryobiopsy

Hemacolor stain is by Merck KGaA, Darmstadt, Germany. Diff-Quik stain is by Sysmex Ltd., Kobe, Japan. * The number includes 75 endoscopically visible lesions. N, number; ROSE, rapid on-site evaluation; TBB, transbronchial biopsy.

## Data Availability

Data are available from the corresponding author upon reasonable request.

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
