# Peer review of "Clinical Utility of Rapid On-Site Evaluation of Touch Imprint Cytology during Cryobiopsy for Peripheral Pulmonary Lesions"

_cancers, 2022, doi:10.3390/cancers14184493_

Round 1
Reviewer 1 Report
This is a retrospective, single center study that investigated the usefulness of ROSE in transbronchial cryobiopsy for diagnosis of lung cancer. The diagnosis by ROSE was compared with the diagnosis using histological samples. The concordance rate wasn’t high (76.2%) due to many cases with false negative. However, there were just 2 cases with false positive, and the positive predictive value was 93.8% and the specificity was 90.0%. Because less cases with false positive is more important than less cases with false positive in ROSE, I agree with the result that ROSE is useful in this study.
My comments were the following.
Comment 1;
What is a bronchoscopist with sufficient experience of pulmonary cytopathology? Please describe the criteria. If a bronchoscopist with sufficient experience of pulmonary cytopathology isn’t a pathologist or cytologist, the concordance between the diagnosis of ROSE by a bronchoscopist with sufficient experience of pulmonary cytopathology and that by pathologist (it might not be rapid on-site). The diagnostic ability has a big impact on the diagnosis of ROSE.
Comment 2;
Were there any differences between the specimens with ROSE positive/histology positive and ROSE negative/histology positive? For example, size of specimen, % of malignant cells in specimen, histology, degree of tumor differentiation etc. If ROSE positive is related with large amounts of malignant cells in specimens, ROSE may be useful to select suitable specimens for molecular diagnosis, such as NGS.
Comment 3:
Did pathologist review the ROSE slides in 2 false positive cases and discuss with the authors? The authors mentioned some possible causes of false positive in discussion. Some of the causes can actually be examined. Please examine and describe it. It is very informative for bronchscopists who do ROSE.
Comment 4:
Did the ROSE results actually affect the number of biopsies in this cohort?
Author Response
This is a retrospective, single center study that investigated the usefulness of ROSE in transbronchial cryobiopsy for diagnosis of lung cancer. The diagnosis by ROSE was compared with the diagnosis using histological samples. The concordance rate wasn’t high (76.2%) due to many cases with false negative. However, there were just 2 cases with false positive, and the positive predictive value was 93.8% and the specificity was 90.0%. Because less cases with false positive is more important than less cases with false positive in ROSE, I agree with the result that ROSE is useful in this study.
Response:
We thank the reviewer for considering our manuscript. Below are our point-by-point responses to all comments, which have helped us substantially improve our manuscript.
Comment 1;
What is a bronchoscopist with sufficient experience of pulmonary cytopathology? Please describe the criteria. If a bronchoscopist with sufficient experience of pulmonary cytopathology isn’t a pathologist or cytologist, the concordance between the diagnosis of ROSE by a bronchoscopist with sufficient experience of pulmonary cytopathology and that by pathologist (it might not be rapid on-site). The diagnostic ability has a big impact on the diagnosis of ROSE.
Response:
We appreciate the reviewer’s concerns on this point. As commented, it is important to describe the criteria of “a bronchoscopist with sufficient experience of pulmonary cytopathology.” Thus, we have revised this sentence in the Material and Methods section. In addition, we have added sentences and table 3 to show the correlation between the results of ROSE-TIC and the cytological results of the first specimen obtained by cryobiopsy. Cytological evaluation by cytotechnologist was not rapid on-site evaluation, and the concordance rate of ROSE-TIC and cytological results was 84.1%.
Revised sentences (red font)
Page 3, lines 127–129
(Before)
Specimens stained with modified Giemsa were immediately evaluated in the same bronchoscopy room by a bronchoscopist with sufficient experience of pulmonary cytopathology (T.I. or T.T.).
(After)
Specimens stained with modified Giemsa were immediately evaluated in the same bronchoscopy room by a bronchoscopist with sufficient experience (at least five years with at least 200 cases per year) of pulmonary cytopathology (T.I. or T.T.).
Added sentences and table 3 (red font)
Page 7, lines 206–213, Page 8, Table 3
(After)
3.4. Correlation between ROSE-TIC and cytological results of the first specimen obtained using cryobiopsy
In total, 38 specimens (60.3%) were judged to be positive for malignancy in cytology. Table 3 shows a comparison of the results of ROSE-TIC with the cytological results of the first specimens obtained using cryobiopsy. The sensitivity, specificity, and positive and negative predictive values of the ROSE-TIC results for cytological results of the corresponding cryobiopsy specimens were 78.9%, 92.0%, 93.8%, and 74.2%, respectively. The concordance rate was 84.1%.
Table 3. Correlation between the results of rapid on-site evaluation of touch imprint cytology and cytological results of the first specimen obtained by cryobiopsy
ROSE-TIC |
Cytological results of the first specimen obtained by cryobiopsy |
||
Malignancy |
Non-malignancy |
Total |
|
Positive |
30 |
2 |
32 |
Negative |
8 |
23 |
31 |
Total |
38 |
25 |
63 |
Sensitivity, 78.9%; specificity, 92.0%; positive predictive value, 93.8%; negative predictive value, 74.2%; concordance rate, 84.1%. ROSE-TIC, rapid on-site evaluation of touch imprint cytology |
Comment 2;
Were there any differences between the specimens with ROSE positive/histology positive and ROSE negative/histology positive? For example, size of specimen, % of malignant cells in specimen, histology, degree of tumor differentiation etc. If ROSE positive is related with large amounts of malignant cells in specimens, ROSE may be useful to select suitable specimens for molecular diagnosis, such as NGS.
Response:
We thank the reviewer for this clinically important comment. Unfortunately, we have not evaluated the size of specimens or % of malignant cells in specimens in this study. As for histology, among 30 specimens with ROSE positive/histology positive, there were 18 adenocarcinomas, 5 metastatic tumors, 4 squamous cell carcinomas, and 3 other tumors. Among the 18 adenocarcinomas, there were 2 well-differentiated, 6 moderate-differentiated, 7 poor-differentiated, and 3 mucinous adenocarcinomas. On the other hand, among 13 specimens with ROSE negative/histology positive, there were 10 adenocarcinomas, 2 metastatic tumors, and a MALT lymphoma. Among the 10 adenocarcinomas, there were 1 well-differentiated, 3 moderate-differentiated, 4 poor-differentiated, and 2 mucinous adenocarcinomas. Thus, we have not found any differences between the specimens with ROSE positive/histology positive and those with ROSE negative/histology positive.
Comment 3:
Did pathologist review the ROSE slides in 2 false positive cases and discuss with the authors? The authors mentioned some possible causes of false positive in discussion. Some of the causes can actually be examined. Please examine and describe it. It is very informative for bronchscopists who do ROSE.
Response:
We appreciate the reviewer’s points. We have reviewed the ROSE slides with a cytology specialist. In one case of adenocarcinoma, an evaluator misjudged degenerative bronchial cells surrounding the lesions as adenocarcinoma. In the other case of lymphoepithelial carcinoma, an evaluator mistook the alveolar epithelium with reactive changes for an adenocarcinoma. In response to this comment, we have added the following sentences in the Results section:
Added sentences (red font)
Page 7, lines 200–203
(After)
In one case of adenocarcinoma, an evaluator misjudged degenerative bronchial cells surrounding the lesions as adenocarcinoma. In the other case of lymphoepithelial carcinoma, an evaluator mistakenly described an alveolar epithelium with reactive changes as an adenocarcinoma.
Comment 4:
Did the ROSE results actually affect the number of biopsies in this cohort?
Response:
We thank the reviewer for this important comment. In this study, the median number of biopsy specimens was 3.5 (range, 1-4) in ROSE positive cases and 3.0 (range, 1-4) in ROSE negative cases (Mann-Whitney’s U test, p = 0.665). Thus, the ROSE results did not affect the number of biopsies in our cohort. A larger, prospective, multi-arm study is required to confirm the clinical benefit of ROSE-TIC during cryobiopsy

Reviewer 2 Report
Mutu et al report on the results of ROSE of criobiopsy specimens from peripheral pulmonary lesions in a retrospective cohort. While the Authors correctly discuss the many limitations of the study, I'd like to add a few criticisms that, if dealt with, might help improve the manuscript.
- Abstract and Conclusions: A concordance rate of 76% means that approximately one fourth of the cases are incorrectly categorized by ROSE, at least when performed by pulmonologists as in this study. These results do not support the conclusions, which I would modify in a more prudent fashion by stressing the above concept.
- Table 3 shows clearly a lower sensitivity and concordance for ROSE of cryobiopsy as compared to ROSE of standard forceps biopsy. Please discuss and provide an hypothesis to explain this finding in the Discussion section (eg, may this be related to the fact that ROSE was performed in the current study by pulmonologists and not by pathologists? other?).
- The Authors compared the ROSE smear from the first cryobiopsy with the corresponding histology specimen. However, as ROSE was performed by a bronchoscopist it would be very useful a comparison between the ROSE smear and the corresponding Papanicolaou-stained smear performed in the pathology dept by a cythotechnologist or a cytopathologist. This would be important because literature shows that, as compared by a pathologist (gold standard), pulmonary lesions are more difficult to categorize by pulmonologists then lymph nodes (please see and cite Natali F. et al. Cythopathology 2020). This comparison would be particularly useful for the 2 false positive cases (how were the Papanicolaou-stained smears of these 2 cases categorised by the cythotechnologist /cytopathologist?).
- In table 1 the unit of measure (median IQR? Mean SD?) should be clearly indicated for each characteristic/parameter.
Author Response
Mutu et al report on the results of ROSE of criobiopsy specimens from peripheral pulmonary lesions in a retrospective cohort. While the Authors correctly discuss the many limitations of the study, I'd like to add a few criticisms that, if dealt with, might help improve the manuscript.
Response:
We thank the reviewer for considering our manuscript. Please find below our point-by-point responses to these comments, which have helped us substantially improve our manuscript.
Comment 1;
Abstract and Conclusions: A concordance rate of 76% means that approximately one fourth of the cases are incorrectly categorized by ROSE, at least when performed by pulmonologists as in this study. These results do not support the conclusions, which I would modify in a more prudent fashion by stressing the above concept.
Response:
We thank the reviewer for this comment. As the reviewer pointed out, it is important that approximately one fourth of the cases were incorrectly categorized by ROSE in our cohort. In response to this comment, we have changed the following sentences in the conclusion:
Revised sentences (red font)
Page 10, lines 311–314
(Before)
Our study suggests that ROSE-TIC has potential utility in cryobiopsy for solid PPLs.
(After)
Although the results of ROSE-TIC performed by bronchoscopists were not consistent with the histological findings of the corresponding cryobiopsy specimens in approximately one fourth of the cases, our study suggests that ROSE-TIC has potential utility in cryobiopsy for solid PPLs.
Comment 2;
Table 3 shows clearly a lower sensitivity and concordance for ROSE of cryobiopsy as compared to ROSE of standard forceps biopsy. Please discuss and provide an hypothesis to explain this finding in the Discussion section (eg, may this be related to the fact that ROSE was performed in the current study by pulmonologists and not by pathologists? other?).
Response:
We thank the reviewer for this comment. As the reviewer pointed out, sensitivity and concordance in ROSE of cryobiopsy seem to be lower than in standard forceps biopsy. Although, to our knowledge, there is no study comparing the results of ROSE-TIC during cryobiopsy performed by trained pulmonologist and pathologists, we believe that this can be explained because ROSE-TIC in our study was performed by pulmonologists. We have added a column titled “Evaluator of ROSE” to Table 4 and also added the following sentences:
Added sentences (red font)
Page 8, lines 237–243
(After)
In addition, as can be seen in Table 4, the sensitivity and concordance rate of ROSE and pathological diagnosis is lower in cryobiopsy than in forceps biopsy. One possible factor might be the evaluator of ROSE. Although a retrospective study with forceps biopsy specimens suggested that ROSE performed by a trained pulmonologist had a high correlation to the formal cytologic report by cytopathologists [25], to our knowledge, there is no study comparing the results of ROSE-TIC during cryobiopsy performed by trained pulmonologists with those performed by pathologists.
Comment 3;
The Authors compared the ROSE smear from the first cryobiopsy with the corresponding histology specimen. However, as ROSE was performed by a bronchoscopist it would be very useful a comparison between the ROSE smear and the corresponding Papanicolaou-stained smear performed in the pathology dept by a cythotechnologist or a cytopathologist. This would be important because literature shows that, as compared by a pathologist (gold standard), pulmonary lesions are more difficult to categorize by pulmonologists then lymph nodes (please see and cite Natali F. et al. Cythopathology 2020). This comparison would be particularly useful for the 2 false positive cases (how were the Papanicolaou-stained smears of these 2 cases categorised by the cythotechnologist /cytopathologist?).
Response:
We thank the reviewer for this clinically important comment. We agree that comparing the ROSE smear and the corresponding Papanicolaou-stained smear performed in the pathology dept by a cytotechnologist and a cytopathologist would be very useful. In addition, as the reviewer pointed out, it should be mentioned that respiratory cytology for pulmonary lesions is difficult to evaluate by pulmonologists. Thus, we have added text and Table 3 to the Result section, and revised sentences in the Discussion section as follows:
Added sentences and table 3 (red font)
Page 7, lines 206–213, Page 8, Table 3
(After)
3.4. Correlation between ROSE-TIC and cytological results of the first specimen obtained using cryobiopsy
In total, 38 specimens (60.3%) were judged to be positive for malignancy in cytology. Table 3 shows a comparison of the results of ROSE-TIC with the cytological results of the first specimens obtained using cryobiopsy. The sensitivity, specificity, and positive and negative predictive values of the ROSE-TIC results for cytological results of the corresponding cryobiopsy specimens were 78.9%, 92.0%, 93.8%, and 74.2%, respectively. The concordance rate was 84.1%.
Table 3. Correlation between the results of rapid on-site evaluation of touch imprint cytology and cytological results of the first specimen obtained by cryobiopsy
ROSE-TIC |
Cytological results of the first specimen obtained by cryobiopsy |
||
Malignancy |
Non-malignancy |
Total |
|
Positive |
30 |
2 |
32 |
Negative |
8 |
23 |
31 |
Total |
38 |
25 |
63 |
Sensitivity, 78.9%; specificity, 92.0%; positive predictive value, 93.8%; negative predictive value, 74.2%; concordance rate, 84.1%. ROSE-TIC, rapid on-site evaluation of touch imprint cytology |
Revised sentences (red font)
Page 9, lines 267–269
(Before)
Respiratory cytology has numerous potential pitfalls.
(After)
Respiratory cytology has numerous potential pitfalls and pulmonary lesions are more difficult to categorize by pulmonologists than lymph nodes [34].
Added references (red font)
Page 12, lines 423–425
(After)
Natali, F.; Cancellieri, A.; Giunchi, F.; Silvestri, A.D.; Livi, V.; Ferrari, M.; Paioli, D.; Betti, S.; Fiorentino, M.; Trisolini, R. Interobserver agreement between pathologist, pulmonologist and molecular pathologist to estimate the tumour burden in rapid on-site evaluation smears from endosonography and guided bronchoscopy. Cytopathology 2020, 31, 303–9, doi:10.1111/cyt.12867.
Comment 4;
In table 1 the unit of measure (median IQR? Mean SD?) should be clearly indicated for each characteristic/parameter.
Response:
We thank the reviewer for this comment. In Table 1, the data are presented as number of patients (%) or median (ranges). We have also described them on the footnote of Table 1.

Round 2
Reviewer 1 Report
The manuscript was rivesed well.
I don't have any additional comments.
Reviewer 2 Report
The Authors have dealt satisfactorily with the criticisms raised in the first round of review.